# Quinolone Resistance of *Actinobacillus pleuropneumoniae* Revealed through Genome and Transcriptome Analyses

**DOI:** 10.3390/ijms221810036

**Published:** 2021-09-17

**Authors:** Xiaoping Ma, Bowen Zheng, Jiafan Wang, Gen Li, Sanjie Cao, Yiping Wen, Xiaobo Huang, Zhicai Zuo, Zhijun Zhong, Yu Gu

**Affiliations:** 1Key Laboratory of Animal Disease and Human Health of Sichuan Province, College of Veterinary Medicine, Sichuan Agricultural University, Chengdu 611130, China; mxp886@sicau.edu.cn (X.M.); zhengbowen@stu.sicau.edu.cn (B.Z.); wangjiafan@stu.sicau.edu.cn (J.W.); 2018303031@stu.sicau.edu.cn (G.L.); zzcjl@126.com (Z.Z.); zhongzhijun488@126.com (Z.Z.); 2Research Center of Swine Disease, College of Veterinary Medicine, Sichuan Agricultural University, Chengdu 611130, China; yueliang5189@163.com (Y.W.); hxb010@126.com (X.H.); 3Bioengineering Department, Sichuan Water Conservancy Vocational College, Chengdu 611231, China; 4College of Life Sciences, Sichuan Agricultural University, Chengdu 611130, China

**Keywords:** *Actinobacillus pleuropneumoniae*, quinolone resistance, transcriptome, porin, efflux pump, quinolone resistance-determining region

## Abstract

*Actinobacillus pleuropneumoniae* is a pathogen that infects pigs and poses a serious threat to the pig industry. The emergence of quinolone-resistant strains of *A.*
*pleuropneumoniae* further limits the choice of treatment. However, the mechanisms behind quinolone resistance in *A.*
*pleuropneumoniae* remain unclear. The genomes of a ciprofloxacin-resistant strain, *A. pleuropneumoniae* SC1810 and its isogenic drug-sensitive counterpart were sequenced and analyzed using various bioinformatics tools, revealing 559 differentially expressed genes. The biological membrane, plasmid-mediated quinolone resistance genes and quinolone resistance-determining region were detected. Upregulated expression of efflux pump genes led to ciprofloxacin resistance. The expression of two porins, OmpP2B and LamB, was significantly downregulated in the mutant. Three nonsynonymous mutations in the mutant strain disrupted the water–metal ion bridge, subsequently reducing the affinity of the quinolone–enzyme complex for metal ions and leading to cross-resistance to multiple quinolones. The mechanism of quinolone resistance in *A. pleuropneumoniae* may involve inhibition of expression of the outer membrane protein genes *ompP2B* and *lamB* to decrease drug influx, overexpression of AcrB in the efflux pump to enhance its drug-pumping ability, and mutation in the quinolone resistance-determining region to weaken the binding of the remaining drugs. These findings will provide new potential targets for treatment.

## 1. Introduction

*Actinobacillus pleuropneumoniae* (APP) is the etiological agent of porcine pleuropneumonia, characterized by serious fibrinous-hemorrhagic necrotizing pneumonia and fibrinous pleurisy. This economically important disease causes considerable losses in the swine industry worldwide [1]. Owing to the diversity of serotypes (19 serotypes) and regional epidemic differences, there is no satisfactory vaccine to control APP infection [2]. Treatment of APP is challenging because of the emergence of drug-resistant phenotypes and various drug resistance mechanisms [3]. Fluoroquinolones are important antibiotics used to treat gram-negative bacterial infections in both human and veterinary medicine [4]; however, therapeutic options against APP are limited by the emergence of quinolone-resistant strains [5,6].

In previous studies, development of quinolone resistance in bacteria was shown to be related to three mechanisms: (i)changes in the affinity of target enzymes and drugs because of chromosomal mutations. Fluoroquinolone resistance is usually associated with mutations in specific regions of gyrase or topoisomerase IV. In general, type II enzyme mutation produces ≤10-fold drug resistance, and higher levels of drug resistance usually appear in strains wherein both enzymes are mutated [7,8]. Quinolones bind to the target enzyme through a water-metal ion bridge formed by a non-catalytic magnesium ion coordinated with four water molecules. Therefore, targeted mutations that lead to drug resistance occur more frequently on amino acid residues that bridge water-metal ions [9,10].(ii)reduced uptake or increased efflux leading to decreased drug accumulation, mainly related to porin and efflux pumps. Porin is located on the outer membrane of bacteria and is mainly composed of β-sheets. When the mass fraction of the drug is less than its exclusion limit, under the action of free diffusion, porin can become the internal flow channel of the drug [11]. Down-regulation, deletion or narrowing of the channel can lead to low-level resistance to quinolones [12,13,14]. Drug efflux is another important means for development of fluoroquinolone resistance. Resistance nodulation division family (RND) is one of the causes for quinolone resistance in gram-negative bacteria [15].(iii)Drug resistance gene carried by plasmids produces target-protective proteins, drug modification enzymes and efflux proteins [12,16]. Additionally, biofilm-positive strains are highly resistant to external antimicrobial agents [17].

Previous studies suggested that quinolone resistance of APP is related to multiple target gene mutations in the quinolone resistance-determining region (QRDR) or an uncharacterized efflux pump as candidate mechanisms [18,19]. However, efflux pumps and influx porins must be further explored, and joint genomic and comparative transcriptomic analysis is needed to provide insight into adaptive resistance evolution [20,21].

In this study, we performed whole-genome sequencing and comparative transcriptome analysis of APP isolates to identify an acquired quinolone resistance mutation. The resistance mechanism of the isolates to quinolones was also evaluated.

## 2. Results

### 2.1. Genome Assembly

Circular chromosomes with a total length of 2,317,956 base pairs (bp) and GC content of 41.2% were assembled (Figure 1a). There were 2,158 open reading frames with a total length of 2,003,163 bp, accounting for 86.42% of the total genome length. A complete circular plasmid with a length of 10,094 bp and GC content of 44.16% was assembled with 14 open-reading frames (Figure 1b).

### 2.2. Transcriptome Results

An average of 2.33 Gbp clean bases was generated from each of the six samples. A total of 2179 genes was assembled and exhibited an average mapping ratio of 96.41%; the average values of Q20 and Q30 were 97.88% and 93.92%, respectively.

Furthermore, sequencing analysis revealed large transcriptome changes in the mutant strains (Figure 2). The correlation of each sample is shown in Appendix A, where little variability was observed between groups. A total of 559 differentially expressed genes (DEGs) was detected in SC1810R compared to in the wild-type (WT) (Appendix A), composed of 201 upregulated and 358 downregulated genes (Appendix A). Gene Ontology (GO) term enrichment analysis of the DEGs was annotated using 186 GO terms, including 87 biological processes, 15 cellular components and 83 molecular functions. Carbohydrate transmembrane transport and phosphoenolpyruvate-dependent sugar phosphotransferase system (PTS) were significantly enriched in biological process and D-glucosamine PTS permease activity, protein-N(PI)-phosphohistidine-sugar phosphotransferase activity and transporter activity in molecular function; for the cellular component, there was no significant enrichment of terms, although the enrichment degree of the outer membrane and membrane were higher (Figure 2a). Four Kyoto Encyclopedia of Genes and Genomes (KEGG) pathways, including ascorbate and aldarate metabolism, PTS, fructose and mannose metabolism and amino sugar and nucleotide sugar metabolism, were significantly enriched in the DEGs (Figure 2b).

### 2.3. Analysis of Bofilm

#### 2.3.1. Biofilm-Forming Capacity of SC1810 and SC1810R

Crystal violet (CV) staining detected strain SC1810 as a strong biofilm producer, whereas the mutant SC1810R exhibited negative biofilm-forming ability (Figure 3a). The effects of ciproflaxin (CPF) on SC1810 biofilm formation and its eradication are shown in Figure 2. At treatment concentrations of 0.0009 to 0.0039 μg/mL, biofilm formation was promoted, whereas treatment with 0.0078 μg/mL CPF significantly inhibited SC1810 biofilm formation and continued its inhibitory effect in a dose-dependent manner (Figure 3b). However, higher concentrations of CPF were required to eradicate the preformed biofilms. Biofilms were significantly eradicated by treatment with CPF at 0.5 μg/mL; even when the CPF concentration was increased to 32 μg/mL, the biofilm biomass was not significantly reduced (Figure 3c).

The effect of nine antimicrobials on the growth of SC1810 and SC1810R was examined (Table 1). Gentamicin and florfenicol showed relatively strong effects on SC1810, increasing the minimum inhibitory concentrations MICs by 1–2 dilutions. SC1810R showed a significantly higher MIC to CPF than SC1810. Enrofloxacin, norfloxacin and levofloxacin showed cross-resistance, with a significantly higher MIC in SC1810R. No differences in susceptibility were observed between SC1810 and SC1810R for doxycycline, erythromycin and sulfamethoxazole.

#### 2.3.2. DEGs in Biofilm 

The genes *pgaABCD* [22] encoding biofilm poly-β-1,6-N-acetylglucosamine synthesis proteins, the regulators *luxS* [23], *vacJ* [24], *clpP* [25], *hns* and *rpoE* on the pga operon associated with APP biofilm formation were not affected in the mutant strain SC1810R. Adhesion is an important stage in APP biofilm formation. Eight genes involved in the adhesion of *A. pleuropneumoniae* were repressed by CPF treatment (Table 2 and Figure 3d). These genes included the type 4 pili genes *apfABCD* [23,26], two genes encoding prepilin peptidase–dependent protein (gene2001, gene2002), *comEA* [27] encoding a DNA uptake and binding protein and type IV pilus secretin gene *pilQ*. Alphatoxins are necessary for the formation of *Staphylococcus aureus* and *Streptococcus pneumoniae* biofilms [28,29]; notably, *apxIIA* encoding alphatoxin, *apxIIC*, which encodes activating lysine acyltransferase, and *tolC2* [30], which contributes to the virulence of *A. pleuropneumoniae*, were downregulated. The quantitative reverse-transcription polymerase chain reaction (qRT-PCR) results showed that *apfA*, *apfC* and *comEA* were significantly downregulated in the mutants (Figure 3e).

### 2.4. Analysis of Porins

Loss, decreased expression and narrow channels of porins also affect the degree of drug resistance. In this case, 12 porins with β-barrel structures were identified in the genome of SC1810; five were downregulated and two were upregulated according to the transcriptome data (Table 3). Porins were marked on a volcanic map (Figure 4a), and the results showed that *ompP2B* (gene680) had the highest fold-change and lowest false discovery rate. To further analyze these DEGs, differences in expression among different porins were compared. A heatmap plot was used to show the gene expression levels of the porins (Figure 4b), and the qRT-PCR results for *ompP2A*, *ompP2B* and *lamB* showed the same trend (Figure 4d). An evolution tree was built and the conserved regions of these porins were predicted (Figure 4c). The results showed that ompP2A and ompP2B, both of which belong to the ompP2 family, have a strong genetic relationship and an almost consistent conservative area. LamB, OmpP2B and OmpP2A are composed of three monomers; bottleneck prediction results showed that the channel sizes of LamB and OmpP2B with downregulated transcription were larger than the others (Table 4). Interestingly, the bottleneck residue of OmpP2A (1.68 Å, upregulated) was smaller than that of OmpP2B (3.54 Å, downregulated), which is the reverse of what was observed in the transcriptomic data.

### 2.5. Analysis of Efflux Pumps

The efflux pump is a factor affecting the degree of drug resistance. A total of 91 efflux pump genes was detected in the genome of SC1810, of which 29 were associated with quinolone resistance (Appendix A). These genes were screened from the DEGs and compared with several databases and analyzed using Swiss-model. Finally, efflux pump genes related to CPF resistance were screened (Table 4). The expression of the efflux pump *macB* (gene663), which was only labeled as associated with macrolide resistance in genome analysis, was upregulated in the transcriptome. Two new unidentified major facilitator super family (MFS)-type efflux pumps (gene835 and gene837) were upregulated in the transcriptional group and were not included in the Comprehensive Antibiotic Resistance Database. We compared the fold-change (Figure 5a) in expression to determine the weight of DEGs. A heatmap plot was used to show gene expression levels (Figure 5b), which indicated that compared with other pumps, AcrA (gene622) and AcrB (gene623) play important roles in CPF resistance.

The relative transcription levels of genes encoding AcrB in the mutant strain (SC1810R) were increased as revealed by qRT-PCR (Figure 5c). HMMER was used to determine that *acrB* (gene623) is the only *acrB*-like RND transporter in the *A. pleuropneumoniae* SC1810 genome with a translated length of 1073 amino acids. A homologous crystal structure of AcrB-*Escherichia coli* (PDB:1T9U) was used as a template to model the homology of AcrB-APP. Docking between the central cavity of AcrB-APP and CPF showed a binding energy of −23.77 KJ/mol (Figure 5d). Residues within 5 Å of bound CPF are shown in Figure 5e, revealing a loose pocket as compared with that of *E. coli* (Figure 5f). The cyclopropyl groups and quinoline carboxylic acid group of CPF form hydrogen bonds with the side chains of Lys48, Asp136, Ser173 and Tyr321 (Figure 5g). Binding appears to involve a larger number of residues, including Phe602 and Phe177, which are homologous residues in the hydrophobic pit in *E. coli* [32]. The hydrophobic pits of *E. coli* are composed of Phe136, Phe178, Phe610, Phe615, Phe617 and Phe628, whereas the hydrophobic pits of APP are composed of Ala137, Phe177, Met597, Phe602 and Ile611.

### 2.6. Analysis of QRDRs

#### 2.6.1. Mutations in QRDRs of SC1810R

Mutations in *gyrA*, *gyrB*, *parC* and *parE* were detected by sequencing. Overall, two substitutions in *gyrA* (S83F, D87N) and one in *parC* (E89k) were identified. No mutations were found in either *gyrB* or *parE*. These results are consistent with previous reports of enrofloxacin-resistant APP strains [18]. G-to-T substitution at position 248 led to a Ser83-Phe substitution, while G-to-A at position 259 led to a Asp97-Asn substitution. In comparison with strain S4074, C-to-T substitution at position 249 in *gyrA* and T-to-G substitution at site 1377 in *parE* already occurred in SC1810 before treatment with CPF. However, in contrast to the results of previous studies, the common D479E substitution was not observed [18].

#### 2.6.2. Structural Comparison of GyrA and ParC

As quinolone–topoisomerase binding is facilitated through a noncatalytic Mg^2+^ coordinating with four water molecules, the local structure and ligand binding of both WT and mutant strains were visualized and compared (Figure 6). For gyrase (Figure 6a,b), the water molecules surrounding the metal ion form hydrogen bonds with Ser83 and Asp87 in the WT strain, whereas these bonds were lost in the mutant strain. In contrast, for topoisomerase IV (Figure 6c,d), the water–metal ion bridge forms hydrogen bonds with Glu89 in the WT strain. The E89K mutation causes the loss of hydrogen bonds in SC1810R.

### 2.7. Plasmid

The results of plasmid assembly and annotation of strain SC1810 showed that the strain did not carry the plasmid-mediated quinolone resistance gene (PMQR), which is consistent with the results of PCR amplification.

## 3. Discussion

### 3.1. CPF Inhibits Biofilm Formation of APP

Bacteria with biofilm-forming capabilities are more adaptable to adverse environments, and the bacterial communities formed are typically difficult to remove by the host, leading to the recurrence of chronic infection [33]. In this study, the biofilm formation ability of the mutant was significantly lower than that of the WT; the factors related to biofilm formation may have been inhibited during the induction of CPF. Even after the environmental pressure was removed, the biofilm-forming ability failed to recover. High concentrations of CPF inhibited and scavenged the biofilm of the WT. CPF not only inhibits the biofilm formation of *S. aureus*, *E. coli* and *Haemophilus influenzae* but also eradicates their biofilms [34,35]. Similar results were observed in this study.

Transcriptome analysis of mutant and WT strains showed that the genes related to adhesion were downregulated. In addition, the alphatoxin gene *apxIIA* and its activating enzyme gene *apxIIC* were inhibited, both of which are necessary for biofilm formation [28]. In a recent study, TolC1 was confirmed to be required for biofilm formation by *A. pleuropneumoniae* without substantial growth inhibition [19]. However, there was no differential expression of *tolC1*, but the virulence-related *tolC2* was significantly downregulated [30]. These results suggest that the effect of CPF on APP biofilms is a complex process and that the biofilm did not promote CPF resistance in the mutant strain.

### 3.2. OmpP2B and LamB Mediate the Influx of CPF

One of the mechanisms of CPF resistance used by gram-negative bacteria includes loss or severe reduction in the number of porins, or mutations leading to reduced permeability of porins [13]. Based on transcriptome analysis, porin LamB- and OmpP2B-encoding genes were downregulated. In previous studies, after deleting the major porin gene *ompK36*, *lamB* was overexpressed to compensate for the loss of the major porin OmpK36 [36]. In the absence of porins, deletion of *lamB* led to decreased sensitivity to carbapenem drugs, indicating that LamB can also be used as a channel for drug influx [36]. Decreased expression of LamB has been observed in isolates with different antibiotic resistance [37] and downregulation of LamB may increase bacterial antibiotic resistance by decreasing intracellular metabolism pathways, which is consistent with our findings [38]. OmpP2 is the channel for the influx of neomycin and β-lactam drugs in *H. influenzae*, which counteracts the efflux of AcrB-Hi (RND family) [11]. Two OmpP2-like porins were obtained from transcriptome and genomic data of *A. pleuropneumoniae* SC1810, and the opposite expression trend of *ompP2A* and *ompP2B* attracted our attention. The bottleneck radius prediction of porin channels showed that among the three trimers, the minimum effective radiuses were estimated as 1.68, 2.25 and 3.54 Å for OmpP2A, LamB and OmpP2B, respectively (Table 3). The hydrophobicity index of the group of residues forming the surface of the layer for the bottleneck of both LamB and OmpP2B was higher than that of OmpP2A, whereas the hydrophobic core that exists within the trimer contributed to the robustness of the trimeric porin [39]. This result suggests that CPF enters cells more easily from LamB and OmpP2B than from OmpP2A, which may explain why the expression of the former was decreased in drug-resistant strains compared to the WT strains.

### 3.3. AcrB Is the Main Efflux Pump Involved in CPF Efflux

Bacteria can pump toxic compounds out of the cell in a process that does not involve drug modification or degradation [13]. Susceptibility to enrofloxacin in the presence of efflux pump inhibitors at 80 mg/mL of Phe-Arg-β-naphthylamide was tested, with the results indicating that efflux pumps contributed to enrofloxacin resistance in *A. pleuropneumoniae* [18]. In addition, TolC1 is related to drug efflux and biofilm formation [19]. TolC belongs to the outer membrane efflux protein family, members of which function in conjunction with three types of transport systems: ATP-binding cassette (ABC), RND and MFS [16]. Among the six efflux pump genes screened from the transcriptome data (Table 4), the RND family member genes *acrA* and *acrB* showed higher fold-changes, and their expression levels in WT strains and mutants were both higher than those of the remaining efflux pumps (Figure 5b). These results suggest that AcrAB is the main efflux pump for CPF.

Several AcrB drug substrates have been shown to bind to two regions in the transporter domain, known as the access pocket and deep binding pocket [40,41]. Simulation of this process showed that CPF was docked to the pocket area of AcrB (Figure 5d–g). The docking results showed that the residues of AcrB form hydrogen bonds with CPF, enabling CPF to be effluxed by acrB in theory.

In addition, the hydrophobic pits of APP’s AcrB pocket differed from those reported in *E. coli* AcrB [32]. It has been reported that AcrB from *A. pleuropneumoniae* and *H. influenzae* are in the same ancient evolutionary cluster, which is far from the new evolutionary cluster containing the AcrB of *E. coli* [11]. The hydrophobic pit of AcrB of *E. coli* is rich in phenylalanine residues, making the site sterically hindered which is beneficial to the export of some aromatic compounds, but limits its potential substrate [11,32].

Drugs can similarly be occluded in the distal binding pocket and not bound firmly to specific residues, oscillating and moving within the voluminous binding pocket, before being pushed out by peristaltic motion [11,42].

### 3.4. QRDR and PMQR

The mutation site in the QRDR region of the CPF-resistant strain found in this study was also found in the enrofloxacin-resistant strain of APP [18]. Target mutations are the major mechanism of quinolone resistance [12]. In our study, amino acid sequence alignment from isolate SC1810 displayed the S83F/D87N mutation in GyrA and E89K in ParC. Structural comparison between the WT and mutant strains showed that amino acid mutation led to the loss of hydrogen bonds in water–metal ion bridges.

Mutations of residues disrupt the water–metal ion interaction and significantly reduce quinolone–enzyme affinity [12]. The mutation at position 83 of GyrA and that at position 85 of ParC may strongly affect drug stability, with the formation of more hydrogen bonds between amino acid residues and the ion bridge of water molecules compared to other sites. Plasmid-mediated quinolone resistance genes were not found in either the WT or mutant strains, indicating that no PMQR mechanism was involved.

## 4. Materials and Methods

### 4.1. Isolates and Growth Conditions

WT APP strain SC1810, isolated from the lungs of a diseased piglet in Sichan China in 2018, was identified as serovar 15 by PCR as previously described [43]. SC1810 and its derivatives were grown in tryptic soy broth (TSB) or on tryptic soy agar (Qingdao Hai Bo Biological Technology Co., Ltd., Zhejiang, China) supplemented with 5% bovine serum (Thermo Fisher Scientific, Waltham, MA, USA) and 10 μg/mL β-nicotinamide adenine dinucleotide (NAD) (Beijing Solarbio Science & Technology Co., Ltd., Beijing, China) at 37 °C and 220 rpm in a shaking incubator. SC1810R (CPF-R), which was derived from the WT strain in this study, was cultured under the same conditions. Briefly, a single colony of SC1810 was cultured in TSB medium with 1/4MIC CPF (Shanghai Sangon Biotech Co., Ltd., Shanghai, China), and the concentration was increased continuously with each passage. When concentration reached 16 μg/mL, it was increased by 8 μg/mL each time, and the strain at each concentration was subcultured twice. The MIC values of the resistant strains were considered meaningful when the strains were continuously cultured in a medium without antibiotics and their MIC values remained unchanged.

### 4.2. Antibiotic Susceptibility Assays

The MICs for quinolone were determined using the broth microdilution method at 37 °C as recommended by the Clinical and Laboratory Standards Institute (CLSI) M100-S29. *Actinobacillus pleuropneumoniae* ATCC 27090 was used for quality control. Susceptibilities were interpreted using the clinical breakpoints established by CLSI M100-S29. All antibiotics were obtained from Sangon Biotech Co., Ltd. The experiments were independently performed at least three times in triplicate.

### 4.3. Genome Sequencing of Strain SC1810

The whole-genome shotgun strategy was adopted to construct the libraries of different inserted fragments, and next-generation sequencing was performed on an Illumina NovaSeq sequencing platform (San Diego, CA, USA). Third-generation single-molecule sequencing technology was used to sequence these libraries on the PacBio Sequel sequencing platform (Menlo Park, CA, USA). The analysis of antibiotic resistance was based on the Comprehensive Antibiotic Resistance Database (https://card.mcmaster.ca/; accessed on 1 July 2021) [44].

### 4.4. RNA-Seq Analysis

#### 4.4.1. RNA Isolation, RNA-Seq Library Construction and Sequencing

Total RNA was isolated from WT and CPF-R cultured in TSB medium (with 0.01% NAD and 5% bovine serum) for 8 h at 37 °C using the Total RNA Isolation System (Promega, Madison, WI, USA) according to the manufacturer’s instructions. RNA degradation and contamination were monitored on 1% agarose gels, and RNA purity (OD_260/280)_ was detected using a NanoPhotometer^®^ spectrophotometer (Implen, Munich, Germany). RNA concentration and integrity were assessed using a Qubit^®^ RNA Assay Kit in Qubit^®^ 2.0 Fluorometer (Life Technologies, Carlsbad, CA, USA) and RNA Nano 6000 Assay Kit of Bioanalyzer 2100 system (Agilent Technologies, Santa Clara, CA, USA), respectively.

Sequencing libraries were generated using rRNA-depleted samples and the NEBNext^®^ Ultra™ Directional RNA library Prep Kit for Illumina^®^ (New England Biolabs, Ipswich, MA, USA) following the manufacturer’s recommendations, and index codes were added to attribute sequences to each sample. Briefly, fragmentation was performed using divalent cations under elevated temperature in NEBNext First Strand Synthesis Reaction Buffer (5×). First-strand cDNA was synthesized using random hexamer primers and M-MuLV reverse transcriptase (RNase H^-^). Second-strand cDNA was subsequently synthesized using DNA Polymerase I and RNase H. Remaining overhangs were converted into blunt ends via exonuclease/polymerase activities.

DNA fragments were ligated with NEBNext Adaptor and amplified with Universal PCR primers and Index (X) Primer for sequencing. PCR products were purified using the AMPure XP system (Beckman Coulter, Brea, CA, USA), and library quality was assessed using an Agilent Bioanalyzer 2100 system (Agilent Technologies).

Index-coded samples were clustered on a cBot Cluster Generation System using Novaseq 6000 PE Cluster Kit (Illumina) according to the manufacturer’s instructions. These library preparations were then sequenced on an Illumina Novaseq 6000 platform, and 150-bp paired-end reads were generated.

#### 4.4.2. Analysis of RNA-Seq Data 

Raw reads generated by high-throughput sequencing were cleaned by removing reads containing adapters. The Q20, Q30 and GC content were calculated to ensure that all downstream analyses were based on high-quality clean data. Reference genome and gene model annotation files were obtained from the previous SC1810-WT strain whole-genome shotgun sequencing as performed earlier [NCBI: CP071698 for chr, CP071699 for plasmid). The index of the reference genome was built using Bowtie v2.0.6 and paired-end clean reads were aligned to the reference genome using TopHat v2.1.1. Differential expression analysis of two conditions/groups (three biological replicates per condition) was performed using the DESeq R package (v1.42.0), which provides statistical routines for determining differential expression in digital gene expression data using a model based on a negative binomial distribution. The resulting *p*-values were adjusted using Benjamini and Hochberg’s approach for controlling the false discovery rate (*q*-value). Genes with an adjusted *q*-value < 0.01, were identified by DESeq and log2 fold-change >1 were assigned as differentially expressed.

#### 4.4.3. And KEGG Pathway Enrichment Analyses of DEGs

GO enrichment analysis of DEGs was performed using the GOseq R package, in which gene length bias was corrected. KOBAS v2.0 was used to test the statistical enrichment of DEGs in KEGG pathways. GO and KEGG terms with corrected *p*-value < 0.05 were considered as significantly enriched by the DEGs.

### 4.5. RNA Extraction and qRT-PCR

Ten genes encoding proteins related to the resistance mechanism of CPF were selected for validation of the RNA sequence results using qRT-PCR (CFX Connect, Bio-Rad, Hercules, CA, USA). Total RNA was extracted from SC1810-WT and SC1810R, as mentioned earlier, and qRT-PCR with SuperReal PreMix SYBR Green (TIANGEN Biotech (Beijing) Co., Ltd., Beijing, China) was performed in triplicate. Relative mRNA levels and expression ratios of the selected genes were normalized to the expression of the 16S rRNA gene, and fold-changes were calculated using the 2^−^^△△Ct^ method [45]. The primers used for qRT-PCR analyses are listed in Appendix A [23,46].

### 4.6. Biofilm Test

#### 4.6.1. Biofilm-Forming Capacity of Isolates

*Actinobacillus pleuropneumoniae* biofilms were grown as previously described [19]. Briefly, an overnight culture of isolates and their derivatives were diluted 1:200 into fresh TSB with 0.01% NAD and 5% bovine serum, and 200-μL aliquots were dispensed per well in microplates (Costar 3599, Corning, NY, USA). After 12 h at 37 °C without shaking, bacterial growth (OD_600_) was measured using a microplate reader (Bio-Rad iMark^TM^ microplate reader). Each well was washed four times before adding 100 μL of CV solution (0.1%, *w/v*), and the cells were incubated for 5 min at room temperature (25 °C). The plate was washed with distilled water after the CV solution was removed, and 100 μL of glacial acetic acid (33%, *v/v*) was added per well to resolve the adherent biofilms. Absorbance was measured at 570 nm using a microplate reader. Data are expressed as the mean ± standard deviation of three experiments.

#### 4.6.2. Biofilm Inhibition and Eradication Assay

Biofilm inhibitory assays were performed as described previously, with minor changes [47]. Briefly, 2 μL of overnight cultures and 198 μL of TSB (supplemented with 0.01% NAD and 5% bovine serum) were dispensed into a 96-well microplate and exposed to different concentrations of CPF. After 12 h at 37 °C, biofilm biomass was detected by CV staining as previously described. For biofilm eradication [47], overnight cultures were diluted 1:200 with the same medium as described above, and 100 μL of bacterial suspension was transferred to each well in the microplates. After 12 h of static incubation at 37 °C, planktonic cells were removed from the supernatant and washed with PBS (pH 7.4), and 200 μL of TSB containing different concentrations of CPF was added to each well. After another 12 h, biofilm biomass was determined using the CV staining method.

### 4.7. Analysis of Porins and Efflux Pumps in Transcriptome Data

The protein sequences of 12 porins were reformulated as a distance matrix of the phylogenetic reconstruction using MEGA-X. The motifs were predicted using MEME [48] and re-mapped with TBTool [49]. Homology models were created using SWISS-MODEL [50]. Docking between AcrB and CPF was modeled using AutoDock [51], and porin bottlenecks were predicted by MOLE v2.0 [52].

### 4.8. Quinolone Resistance Mutation Identification

To identify the QRDR mutations in the mutant strains, *gyrA*, *gyrB*, *parC* and *parE* were amplified and sequenced using the primers listed in Appendix A [18]. Some primers were constructed using *A. pleuropneumoniae* S4074 gene sequence data (GenBank accession number NZ_CP030753.1). PCR was performed in 50-μL volumes consisting of 2.5 U of TaKaRa *Taq* DNA polymerase, 200 μM of dNTPs, 1.5 mM MgCl_2_ and 50 pmol of each primer in a Bio-Rad T100^TM^ Thermal Cycler for 33 cycles of denaturation at 94 °C for 30 s, annealing at 55–60 °C for 30 s and extension at 72 °C for 1 min, and final extension at 72 °C for 10 min. PCR-amplified products were purified with the QIAquick PCR Purification Kit, following the manufacturer’s recommendations for sequencing. Sequencing was performed using an ABI prism 3730 Automated DNA Analyzer (Applied Biosystems, Foster City, CA, USA). Sequence analysis was carried out by multiple sequence alignment using DNAMAN and MEGA-X. Homology models were created using SWISS-MODEL [50].

### 4.9. PCR Detection of Plasmid-Mediated Resistance Genes

PCR was conducted to simultaneously detect *qnrA*, *qnrB*, *qnrC*, *qnrD*, *qnrS*, *qepA* and *aac(6′)-Ib-cr* genes using specific primers previously reported for *E. coli* and *S. enterica* [53,54].

## 5. Conclusions

We present the drug resistance mechanism for APP SC1810R, as shown in Figure 7. In summary, the mutant lost its biofilm-forming ability and no PMQR was detected, indicating that biofilm and PMQR did not contribute to CPF resistance. Based on our transcriptome data and structural properties of porins, OmpP2B may be the major outer membrane porin protein of APP, which in combination with maltoporin LamB mediates the cellular influx of CPF. The expression of *ompP2B* and *lamB* was decreased to reduce intracellular drug influx in CPF-resistant strains, and the efflux system AcrAB–TolC effluxes the residual CPF in the cytoplasm and periplasm. Mutations in amino acid residues at positions 83 and 87 in GyrA and at position 89 in ParC were among the major causes of CPF resistance in strain SC1810R.

## Figures and Tables

**Figure 1 ijms-22-10036-f001:**
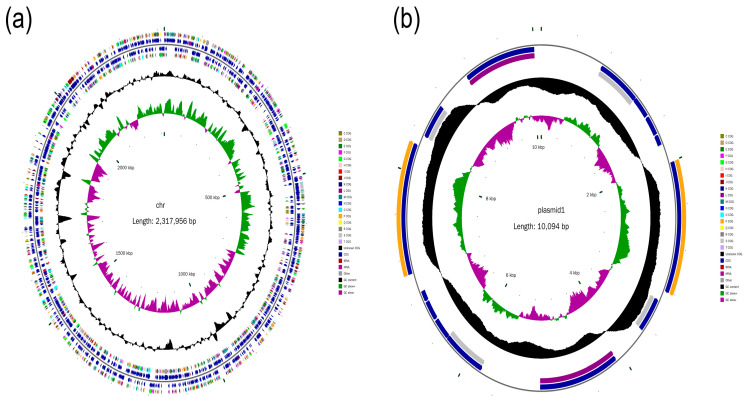
Chromosome genome and plasmid of APP SC1810. (**a**) Chromosome circle map (**b**) Plasmid circle map. From the inside out, the first circle represents the scale; second circle represents the GCSkew; third circle represents the GC content; fourth and seventh circles represent the fifth and sixth circles of COG; to which each coding sequence belongs, representing the position of coding sequence, tRNA and rRNA on the genome.

**Figure 2 ijms-22-10036-f002:**
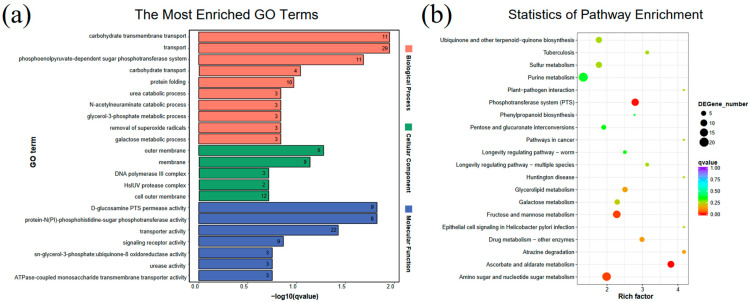
RNA-sequencing analysis. (**a**) GO enriched analysis. Number on the bar chart indicates the number of genes enriched under the term (**b**) and KEGG pathway enriched analysis for up-regulated and down-regulated genes.

**Figure 3 ijms-22-10036-f003:**
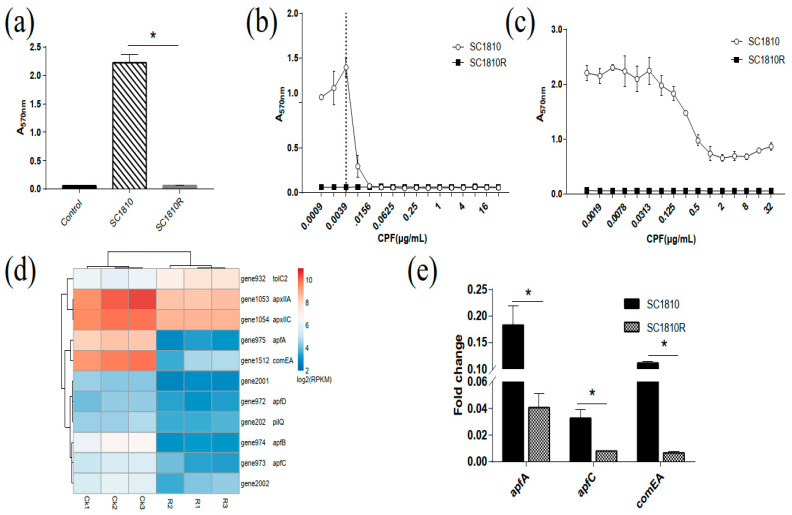
Antibiofilm activity of CPF on SC1810. (**a**) Biofilm biomass of wild-type and mutant strain (* *p* < 0.05). (**b**) Effects of CPF on biofilm formation (12 h with drug, *n* = 3) and (**c**) eradication at different concentrations (12 h without drug, 12 h with drug, *n* = 3), respectively. The data are shown as the absorbance at 570 nm (A570 nm) of residual biofilm and associated error bars denote the standard error of the mean. (**d**) Analysis of differentially expressed genes (DEGs) related to pili formation between wild-type strains and mutants. (**e**) Relative transcription levels of *apfA*, *apfC* and *comEA* determined using qRT-PCR. Data are expressed as the mean and standard deviation (SD) of 3 experiments (* *p* < 0.05).

**Figure 4 ijms-22-10036-f004:**
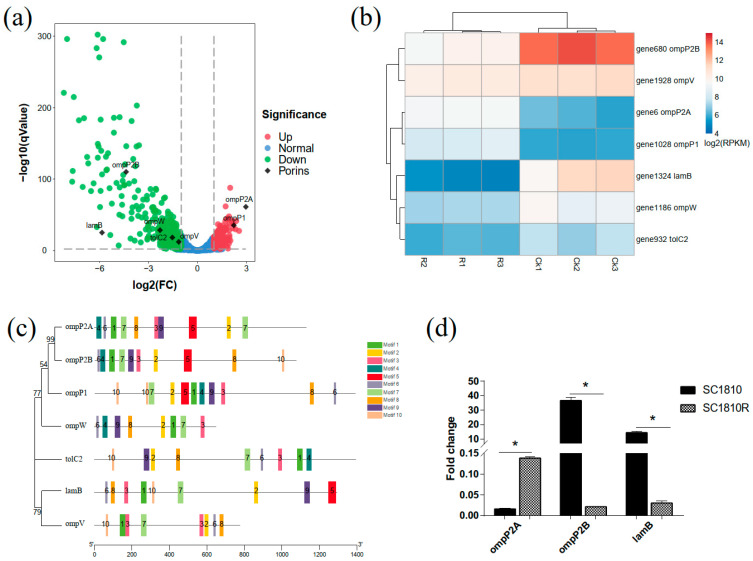
Analysis of differential expression of porins. (**a**) Volcano plots marked with DEGs of porins. (**b**) Each row represents the relative expression of single transcript, and each column represents a sample. Colors represent the log_2_-transformed gene expression level, with red and blue representing high and low expression levels, respectively. (**c**) Molecular phylogenetic analysis by maximum likelihood method and motif prediction. (i) The evolutionary history was inferred by using the maximum likelihood method based on the Tamura 3-parameter model. The percentage of trees in which the associated taxa clustered together is shown next to the branches. (ii) Motifs prediction. The same color or number represents the same conservative site. (**d**) Relative transcription levels of *ompP2A*, *ompP2B* and *lamB*. Data are expressed as mean and standard deviation (SD) of 3 experiments (* *p* < 0.05).

**Figure 5 ijms-22-10036-f005:**
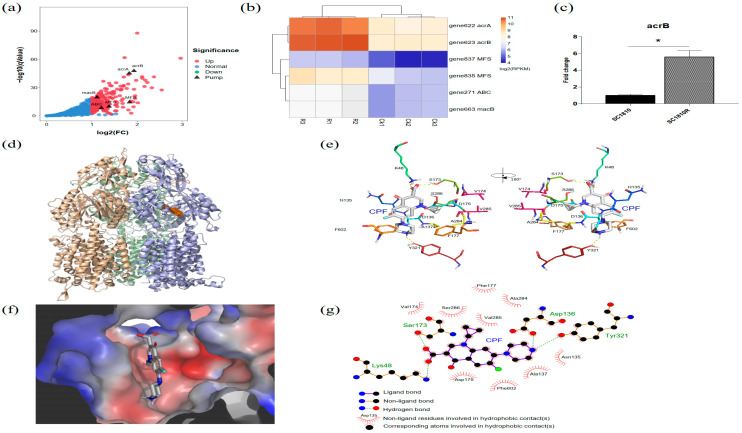
Analysis of differential expression genes of efflux pump (**a**–**c**) and prediction of crystal structure of CPF-bound AcrB (**d**–**g**). (**a**) Volcano plots marked with DEGs of pumps. (**b**) Each row represents the relative expression of single transcript, and each column represents a sample. Colors represent the log_2_-transformed gene expression level, with red and blue representing high and low expression levels, respectively. (**c**) Relative transcription levels of *acrB*. The data are expressed as the mean and standard deviation (SD) of 3 experiments (* *p* < 0.05). (**d**) Whole trimer structure of AcrB (ribbon model) bound with CPF (electron density map). (**e**) Close-up view of the CPF-binding site. Electron density of CPF (orange mesh) overlapped with stick models of LMNG (grey). (**f**) View of the bound CPF (grey), shown in the cut view of the surface model of the distal pocket in the binding monomer. The red indicates higher electron cloud density (gain electrons), whereas blue represents lower electron cloud density (electron loss). (**g**) 2D representation of the interaction between CPF and AcrB was drawn using LigPlot+ [31].

**Figure 6 ijms-22-10036-f006:**
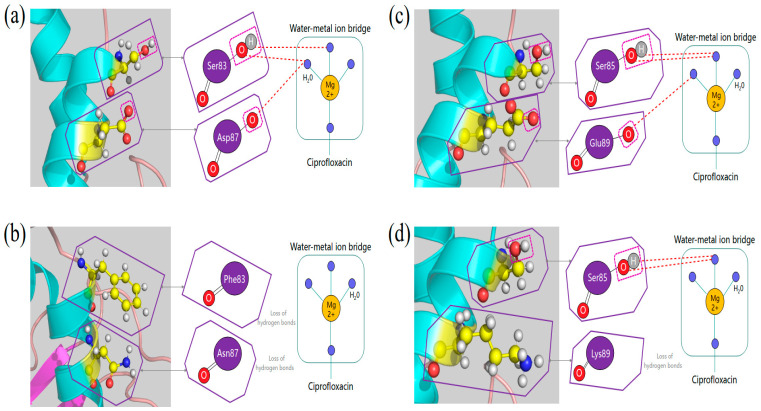
Structural comparison of GyrA S83F/D87N and ParC E89K mutation between wild and mutant strains. (**a**) Quinolone and gyrase binding in the wild-type strain is facilitated by a water-metal ion bridge, which forms hydrogen bonds between the water molecules and Ser 83/Asp87 amino acid. (**b**) Hydrogen bonds are missing in the mutant APP strain, decreasing the binding affinity of quinolone. (**c**) Quinolone and topoisomerase IV binding in the wild-type strain occurs through a water-metal ion bridge, where forms a hydrogen bond between the quinolone and Ser 85/Glu89 amino acid that act as anchor points to enzyme. (**d**) Hydrogen bond link with 89 amino acid is missing, weakening the binding affinity of quinolone.

**Figure 7 ijms-22-10036-f007:**
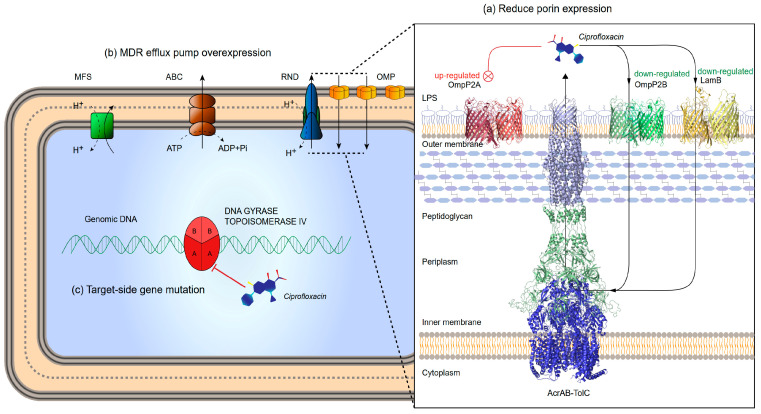
Mechanism of quinolone resistance of *A. pleuropneumoniae*. (**a**) Expression of major porins ompP2B and LamB was decreased, which reduced the drug concentration entering the periplasm. (**b**) Overexpression of multidrug resistance efflux pump reduced the intracellular drug concentraTable 1. RNA sequencing analysis. (**a**) Pearson correlation of all sample. (**b**) Heatmap showing gene expression patterns. (**c**) Volcano plots shows the difference of expression level between the wild and mutant strains.

**Table 1 ijms-22-10036-t001:** Susceptibility of *A. pleuropneumoniae* strains to different antimicrobials.

Antimicrobial Agent	MIC (μg/mL)	
	SC1810	SC1810R
Enrofloxacin	0.125	32
Norfloxacin	0.3125	128
Levofloxacin	0.0039	16
Ciprofloxacin	0.0039	32
Doxycycline	16	16
Gentamicin	0.5	2
Erythromycin	2	2
Florfenicol	64	128
Sulfamethoxazole	>128	>128

**Table 2 ijms-22-10036-t002:** DEGs related to biofilm formation.

Gene ID	Gene	Description	Log2FC
gene975	*apfA*	Possible prepilin peptidase dependent protein D	−5.49
gene974	*apfB*	Protein transport protein HofB-like protein, Pili/fimbriae biogenesis protein	−3.61
gene973	*apfC*	Protein transport protein HofC-like protein	−2.05
gene972	*apfD*	Type IV prepillin dependent peptidase	−1.36
gene202	*pilQ*	Type IV pilus secretin PilQ	−1.26
gene2002	*-*	prepilin peptidase dependent protein A	−1.98
gene2001	*-*	prepilin peptidase dependent protein B	−1.82
gene1512	*comEA*	DNA uptake protein and related DNA-binding protein	−5.39
gene1053	*apxIIA*	RTX-II toxin determinant A	−1.84
gene1054	*apxIIC*	RTX-II toxin-activating lysine-acyltransferase ApxIIC	−1.33
gene932	*tolC2*	TolC family protein	−1.54

**Table 3 ijms-22-10036-t003:** DEGs and structural properties of porins.

Gene ID	Name	Rad (Å)	Hdry	Hdp	Pol	Residues	log2FC
gene6	OmpP2A	1.68	−2.93	−0.54	34.96	R40/K42/N149	2.96
gene1324	LamB	2.25	−2.03	0.39	17.64	Y142/D140/Y29	−5.85
gene680	OmpP2B	3.54	−1.5	0.32	34.71	R70/R37/I144	−4.38
gene1028	OmpP1	0.98	−1.03	0.18	2.13	Y40/N43/N55/V31	2.23
gene1186	OmpW	1.17	2.37	1.33	0.79	T178/Le77/V206	−2.3
gene1928	OmpV	1.46	−1.73	−0.15	2.79	Y255/G224/N212	−1.15
gene932	TolC2	6.77	−0.14	0.3	1.76	Q318c/L115c/N114c/Y129c/L339b	−1.54

Rad: Minimum tunnel radius along that specific layer; Hdry: Average hydropathy index of group of residues making up the surface of the tunnel layer; Hdp: Average hydrophobicity index of group of residues making up the surface of the layer; Pol: Average polarity index of the group of residues making up the surface of the layer; Residues: Group of residues making up the surface of the layer.

**Table 4 ijms-22-10036-t004:** Overexpression efflux pump of transcriptome.

Gene ID	Pump Family	Description	Log2FC
gene835	MFS	hypothetical protein	1.83
gene837		MFS transporter YcaD	1.38
gene663	ABC	MacB family efflux pump subunit	1.11
gene271		ABC transporter permease	1.21
gene622	RND	acrA-like transporter periplasmic adaptor	1.83
gene623		acrB-like transporter	1.93

## Data Availability

All raw reads of samples are available from the National Center for Biotechnology information (NCBI) database. The sequencing data for SC1810 genome has been submitted to GenBank with BioProject record number PRJNA712930. RNA-sequencing data are available from the NCBI short read archive under GEO Accession GSE181030.

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
