# Peer review of "Quinolone Resistance of Actinobacillus pleuropneumoniae Revealed through Genome and Transcriptome Analyses"

_ijms, 2021, doi:10.3390/ijms221810036_

Round 1

Reviewer 1 Report

Quinolone Resistance of Actinobacillus pleuropneumoniae revealed through genome and transcriptome analyses

The objective of the study was to discover an acquired quinolone resistance mutation of Actinobacillus pleuropneumoniae (APP) strains using whole-genome sequencing and comparative transcripome analysis of APP isolates. Porcine pleuropneumonia a disease caused by APP is economically important disease conducting to considerable losses in the swine industry worldwide. Fluoroquinolones are important antibiotics used to treat infection caused by gram -negative bacteria including APP. However, therapeutic options against APP are limited by the emergence of quinolone-resistant strains. Thus identify the resistance mechanism of the APP isolates is really important.

Generally contents of all sections are appropriate and adequate. Materials and methods used in the study are adequately described. Tables and figures are well-presented. Results are generally well described and presented in the manuscript as well as discussion which is comprehensive.. I made only some minor comments listed below.

Minor comments:

Line 403: Add the kit name used to qRT-PCR.

Lines 115-116: Please check if the sentence it correct. In my opinion should be “SC1810R”.

Lines 335-336: Could you provide more detailed description how SC1810R was derived from the WT?

Author Response

Minor comments:

Line 403: Add the kit name used to qRT-PCR.

Response:We have added the name of the kit (section 4.5 RNA Extraction and qRT-PCR; page 13, line 436).

Lines 115-116: Please check if the sentence it correct. In my opinion should be “SC1810R”.

Response: We apologize for this error. We have corrected it (page 5, line 139).

Lines 335-336: Could you provide more detailed description how SC1810R was derived from the WT?

Response: Thank you for your suggestion. We have added the details regarding the induction of drug resistance of strain SC1810R (section 4.1. Isolates and Growth Conditions; page 11, lines 358–365).

Reviewer 2 Report

The manuscript entitled “Quinolone Resistance of Actinobacillus pleuropneumoniae 2 Revealed Through Genome and Transcriptome Analyses” is of considerable topicality and importance because it deals with a topic of increasing popularity within the extremely complex framework of antimicrobial resistance.

The whole manuscript is well structured and the scientific rigor with which the Authors have carried out the experimental plan and the methods used to obtain their results is unexceptionable.

However, the "introduction" section is very poor and could be enriched both with some more detailed information about the main actor of the investigation, Actinobacillus pleuropneumoniae, and with some more reference about the mechanisms of acquisition of bacterial resistance and of some bacilli to the class of quinolones.

For all the other parts, especially the discussion and the conclusions, the Authors have argued their work in an impeccable way and I congratulate them for the excellent research carried out that absolutely must be published as soon as possible.

Author Response

Response:Thank you for your valuable suggestions. We have supplemented our manuscript with some details related to APP (Introduction section; page 1, lines 40–41) and added corresponding background information on the drug resistance mechanism, which was the focus of the study (Introduction section; page 2, lines 50–57 and 59–67).